# Effect of Novel Antibacterial Composites on Bacterial Biofilms

**DOI:** 10.3390/jfb11030055

**Published:** 2020-08-01

**Authors:** Rayan B. Yaghmoor, Wendy Xia, Paul Ashley, Elaine Allan, Anne M. Young

**Affiliations:** 1Department of Biomaterials and Tissue Engineering/Department of Microbial Diseases, UCL Eastman Dental Institute, London, NW3 2QG, UK; rayan.yaghmoor.18@ucl.ac.uk; 2Department of Restorative Dentistry, Umm Al-Qura University, College of Dental Medicine, Makkah 24381, Saudi Arabia; 3Department of Biomaterials and Tissue Engineering, UCL Eastman Dental Institute, London NW3 2QG, UK; wendy.xia2011@yahoo.co.uk; 4Unit of Paediatric Dentistry, UCL Eastman Dental Institute, London WC1E 6DE, UK; p.ashley@ucl.ac.uk; 5Department of Microbial Diseases, UCL Eastman Dental Institute, London NW3 2QG, UK; e.allan@ucl.ac.uk

**Keywords:** dental composite, antibacterial, antibiofilm, polylysine

## Abstract

Continuing cariogenic bacterial growth demineralizing dentine beneath a composite filling is the most common cause of tooth restoration failure. Novel composites with antibacterial polylysine (PLS) (0, 4, 6, or 8 wt%) in its filler phase were therefore produced. Remineralising monocalcium phosphate was also included at double the PLS weight. Antibacterial studies involved set composite disc placement in 1% sucrose-supplemented broth containing *Streptococcus mutans* (UA159). Relative surface bacterial biofilm mass (*n* = 4) after 24 h was determined by crystal violet-binding. Live/dead bacteria and biofilm thickness (*n* = 3) were assessed using confocal laser scanning microscopy (CLSM). To understand results and model possible in vivo benefits, cumulative PLS release from discs into water (*n* = 3) was determined by a ninhydrin assay. Results showed biofilm mass and thickness decreased linearly by 28% and 33%, respectively, upon increasing PLS from 0% to 8%. With 4, 6, and 8 wt% PLS, respectively, biofilm dead bacterial percentages and PLS release at 24 h were 20%, 60%, and 80% and 85, 163, and 241 μg/disc. Furthermore, initial PLS release was proportional to the square root of time and levelled after 1, 2, and 3 months at 13%, 28%, and 42%. This suggested diffusion controlled release from water-exposed composite surface layers of 65, 140, and 210 μm thickness, respectively. In conclusion, increasing PLS release initially in any gaps under the restoration to kill residual bacteria or longer-term following composite/tooth interface damage might help prevent recurrent caries.

## 1. Introduction

Dental caries is a main public health issue worldwide [1]. In Europe, it affects almost 100% of adults and 20–90% of 6-year-old children [2]. Mercury silver amalgam was for nearly 200 years the most commonly used permanent direct dental filling material in the EU. European legislation, however, in July 2018, aligned with the Minamata convention on mercury reduction to ban the use of dental amalgam for children under 15 years old and pregnant or breastfeeding women. Complete dental amalgam phase out is planned for 2030 [3]. Mercury concerns in combination with growing demand for more conservative and aesthetic restorations have made the resin composite the new main direct long-term restorative of choice [4,5]. Composites, however, have higher failure rate with the most common cause being recurrent (secondary) caries [5,6,7,8]. This is largely due to the tendency of resin composite to accumulate bacterial biofilms more than amalgam, which is not only responsible for recurrent caries but also periodontal problems [9].

Dental plaque biofilms consist of microorganisms in an exopolymer matrix (EPM) [10]. These biofilms are three dimensional, spatially-organized microbial communities that show properties as a unit [11]. Metagenomics studies revealed that up to 19,000 bacterial species could be found in a mature dental plaque biofilm [12,13]. Shift in composition of the dental biofilm towards more acidogenic and aciduric bacteria leads to caries. Dentinal caries lesions contain primarily acid-producing and Gram-positive species, with *Streptococcus mutans* being particularly high [14,15]. Their lowering of pH enhances solubility and dissolution of apatite in enamel [16]. Subsequent demineralization of dentine activates enzymes that, over time, slowly hydrolyze remaining collagen [16].

Previously, tooth restoration involved removal of the soft, highly-infected as well as the underlying harder, acid-affected dentine through drilling. Tooth restoration became a major problem during the global SARS-Cov-2 pandemic in 2020, as drilling increases aerosol production that could spread the viral infection [17,18]. Additionally, acid-affected dentine is remineralizable [19]. Furthermore, current recommendations for management of cavitated caries advocate controlling the lesion [19]. This could be better achieved through minimally invasive treatment and bacterial inactivation instead of total demineralized dentine removal [19]. This would reduce the need for drilling, which both dentists and patients consider a major benefit [20].

Bacterial inactivation by composite restorations can be achieved by cavity sealing reducing nutrient ingress [19]. Composite shrinkage during placement or long-term cyclical loading, however, can damage this seal. Residual bacteria or bacterial leakage from surface plaque with sugars from food would then enable biofilm formation in gaps at the composite/tooth interface. Immediate release of antibacterial agents in response to interface damage could potentially reduce this risk. In combination with remineralising agents to seal any gaps, this could provide the time required for the tooth to self-heal.

(Poly-ε-lysine) (ε-PL) (PLS) is a naturally created antibacterial homopolypeptide of 25–30 L-lysine residues that is stable in basic and acidic environments as well as at high temperatures [21,22]. Moreover, it is biodegradable, water-soluble, edible, non-toxic, and has broad-spectrum antibacterial properties [21,23]. Polylysine has therefore recently been incorporated into resin composites with remineralising agents [22,24]. Chemical setting kinetics, volumetric stability, polylysine release kinetics, remineralising features, mechanical properties, and wear have already been investigated [22,24]. One recent study has also studied material effects on bacteria [25]. This previous investigation, however, was restricted to formulations with low PLS concentrations and planktonic *S. mutans* with none of the sucrose required for biofilm maturation [25]. Biofilms, in general, have been shown to be more resistant to antimicrobial agents compared to the same bacterium growing in planktonic culture [26].

The aim of this new study was to evaluate if higher levels of PLS and remineralising agents within composites have the potential to prevent residual bacteria in cavities depositing and forming biofilms early on the composite surface. This is achieved through investigation of new composites incubated with *S. mutans* and sucrose. PLS release kinetics are also provided to help explain the observations.

## 2. Materials and Methods

### 2.1. Composite Components

Resin-composite formulations were prepared as in previous studies [25] but with higher antibacterial and remineralising agents by combining two phases: liquid and powder. Liquid components included urethane dimethacrylate (UDMA) as the main base monomer and the initiator camphorquinone (CQ). Both were from DMG (Hamburg, Germany). Poly (propylene glycol) dimethacrylate (PPGDMA) was added to improve fluidity and polymerization kinetics. This and the adhesion promoting monomer, 4-methacryloyloxyethyl trimellitic anhydride (4-META), were from Polysciences (PA, Warrington, UK) [25].

The filler consisted primarily of silane-treated radiopaque strontium aluminosilicate glass filler of 7 and 0.7 µm from DMG. These were combined with silica nano-glass particles (Azelis, Hertford, UK). Monocalcium phosphate monohydrate (MCPM) (Himed, Old Bethpage, NY, USA) and polylysine (PLS) (Handary, Brussels, Belgium) were added as remineralising and antibacterial agents, respectively. Particle SEM images are provided in a previous work [24].

### 2.2. Paste and Solid Sample Disc Composition and Preparation

Four semi-fluid paste formulations were prepared using a powder to liquid weight ratio of 3:1. The clear liquid phase consisted of UDMA (72 wt%), PPGDMA, (24 wt%), 4-META (3 wt%), and CQ (1 wt%). This monomer system polymerizes effectively upon light activation without need of an amine activator [24]. Then, 7 µm glass, 0.7 µm glass, and silica were combined in the weight ratio 6:3:1 [24]. This maximized particle packing and provided flow and material handling characteristics approved by clinical focus groups. MCPM (0, 8, 12, and 16 wt%) and PLS (0, 4, 6, and 8 wt%) were added to this powder with the MCPM:PLS weight ratio fixed at 2:1.

Clear liquids were prepared by completely dissolving solid 4-META and CQ in PPGDMA followed by UDMA addition. The powder components were mixed first at 3500 rpm for 10 s using a Speedmixer (DAC600.2 CM51, Synergy Devices Ltd., High Wycombe, UK). Monomers and powders were then mixed at 3500 rpm for 40 s.

To prepare solid disc specimens (1 mm thick, 9.4 mm diameter, and mass 0.13 g), pastes were pressed within metal circlips sandwiched between two acetate sheets then light cured. Each surface was cured using a light-emitting diode (LED) with a wavelength of 450–470 nm and power of 1100–1300 mW/cm^2^ (Demi Plus, Kerr Dental, Bioggio, Switzerland) in contact with the acetate sheet with 4 overlapping irradiation cycles according to ISO:4049 [27]. Then, 40 s per cycle was used to ensure maximum room temperature monomer conversion of 72% [24]. After curing, composite discs were removed from the moulds, and excess composite was trimmed.

### 2.3. Biofilm Formation on Composite Discs

*S. mutans* (UA159) was cultured on brain heart infusion (BHI) Agar (CM1136 by OXOID, Thermo Fisher Scientific, Loughborough, UK) at 37 °C in an atmosphere of air enriched with 5% CO_2_ for 72 h. A single colony was inoculated into 10 mL BHI broth (CM1135 by OXOID) and allowed to grow for 18 h at 37 °C in air enriched with 5% CO_2_. This was diluted to obtain a bacterial density of 8 × 10^7^ CFU/mL, which was confirmed by colony forming unit (CFU) counting and calibrated optical density (caused by absorbance and light scatter) at 595 nm (OD_595_). The OD_595_ was measured using a spectrophotometer (Biochrom WPA CO8000, Cambridge, UK).

Specimen discs were placed randomly in 24 well plates (Greiner bio-one, Cellstar, Kremsmunster, Austria) and sterilized through top and bottom surface exposure to UV light for 40 min. Biofilms were grown on each composite disc by adding 25 μL of 8 × 10^7^ CFU/mL inoculum (containing 2 × 10^6^ CFU) with 1 mL BHI broth (OXOID) containing 1% (*w*/*v*) sucrose. The well plate was incubated at 37 °C statically in air enriched with 5% CO_2_. After 24 h, the medium was carefully removed. The resultant biofilms were washed twice with 1 mL of sterile phosphate buffered saline (PBS) and fixed by exposure to 1 mL methanol during 15 min.

### 2.4. Crystal Violet (CV) Biofilm Assay

A crystal violet (CV) assay was used to evaluate relative biofilm mass on composite surfaces [28]. Biofilms were produced on 4 separate days; 3 composite discs per formula were tested each day requiring 12 discs per formula in total.

The procedure involved addition of 1 mL of aqueous 0.1% CV (Pro-Lab Diagnostics, Bromborough, UK) to each disc for 5 min then washing with 1 mL of PBS twice. After drying, bound CV was solubilized in 1 mL of 30% aqueous acetic acid. Once fully dissolved, the resultant CV solutions were diluted to enable absorbance (OD_595_) determination, and the OD_595_ of the undiluted solutions were calculated and reported.

### 2.5. Live/Dead and Biofilm Thickness Assessment

Live/dead assay and biofilm thickness were assessed using confocal laser scanning microscopy (CLSM, Lasersharp 2000) (Radiance 2100, Biorad, Hercules, CA, USA) with a 40× magnification wet lens. Biofilms were prepared and examined on different days in 3 independent assays with one disc per formulation each day.

Then, 20 µL of LIVE/DEAD stain (Viability Kit, Thermofisher Scientific, Loughborough, UK) and 10 mL of PBS were added to the biofilms on discs within wells. Green and red images of live versus dead bacteria were obtained from near the biofilm centre (260 × 260 µm^2^). In the following, red and green images are shown overlapped. For analysis, the individual red or green images were converted to black (background) and white (stained bacteria), and the percentage area that was white was determined using ImageJ software. The ratio of live/dead bacteria was obtained by dividing the percentage of white areas from the green versus red images; this was then converted to percentages. Additionally, movement of the microscope stage in the z direction enabled determination of biofilm thickness. Average results at three random points within the image area were obtained. Normalised thickness was obtained by dividing biofilm thickness by that for the control with no PLS on each given day.

### 2.6. Polylysine Release

To quantify kinetics of PLS release, 6 composite discs of each formulation of known total mass were stored in plastic pots containing 5 mL of deionized water at room temperature (RT). A total of 2 discs per pot enabled sufficient release for ease of detection and a sample repetition number of *n* = 3. The discs were transferred to new plastic containers with fresh 5 mL of deionized water at the following time points: 6 h, 24 h, 3 days, and at 1, 2, 3, 4, 5, 6, 7, 8, 9, 10, 12, 14, and 18 weeks.

The concentration of PLS that had leached into the deionized water was determined using a ninhydrin assay that was modified from previously published work [29]. It involved adding 1 mL of 0.5% ethanolic Ninhydrin (Sigma-Aldrich, Gillingham, UK) solution to boiling tubes containing 4 mL of sample storage solution or aqueous polylysine solutions of known concentration. After vortex-mixing for 10 s, 1.5 g (± 0.1) of marble chips (BDH, Poole, England, UK) were added to each tube. All tubes were covered with aluminium foil and placed in a boiling water bath to accelerate a ninhydrin/PLS reaction. After 15 min, reactions were quenched by rapid cooling to RT.

Following sedimentation, supernatant liquids were analysed using a spectrophotometer (Thermo Spectronic Unicam UV 500 Spectrophotometer, Loughborough, England with Vision pro^TM^ software). The absorbance was recorded between 400–700 nm to ensure curves were consistent and free of background scattering (for example due to residual non sedimented marble chips). Absorbances at 570 nm (A_570_) for known polylysine concentrations of 0, 4, 10, 16, 20, 40, 60, 80, and 100 ppm were used to establish a calibration graph (A_570_ = 0.023/ppm, Pearson correlation coefficient, RSQ = 0.96). This was then used to determine polylysine concentration in each storage solution and cumulative release in μg/disc versus time. Cumulative PLS (%) was also calculated using:(1)$$\%\;\mathrm{PLS}\;\mathrm{release}=\frac{100[\sum_{0}^{t}R_{t}]}{w_{c}}$$
where $R_{t}$ is the amount of released PLS (g) at time *t*, $w_{c}$ is the amount of *PLS* (g) in the composite disc.

### 2.7. Statistical Analysis

The 95% confidence intervals (95%CI) (equal to 2 SD/SQRT(*n*) where SD and SQRT are standard deviation and square root) are provided as error bars in Figures. Average percentage standard deviations (%SD) were 7%, 9%, and 15% and sample numbers *n* = 4, 3, and 3 for crystal violet (CV) absorbance, normalised biofilm thickness, and PLS release studies, respectively. Significance (*p* < 0.05) was determined using one-way Analysis of Variance (ANOVA) followed by pairwise comparisons (Kruskal–Wallis test). Excel was used for curve fitting and to calculate the Pearson correlation coefficient (RSQ).

## 3. Results

### 3.1. Biofilm Mass

CV adsorption and subsequent absorbance indicated an inverse relationship between the amount of biofilm mass and the concentration of PLS in composite formulations (Figure 1). All PLS-containing formulations (4, 6, and 8 wt%) had biofilms with significantly less CV adsorption than the control formula with no PLS. In addition, the formula containing 8% PLS had significantly less CV adsorption compared to the material containing 4% PLS.

### 3.2. Live/Dead S.Mutans

Confocal microscopy images showed primarily dead bacteria with 8% PLS but live bacteria with 0% and 4% PLS levels. The 6% PLS samples had both live and dead bacteria (Figure 2). Quantification of live versus dead stain confirmed that formulae containing 6% and 8% PLS had significantly more dead *S. mutans* and significantly less live *S. mutans* than the 4% and the control formulations (Figure 3). Additionally, live and dead percentages gave linear trends versus concentration between 4% and 8% PLS (Figure 3). Additionally, live and dead percentages gave linear trends versus concentration between 4% and 8% PLS (Figure 3).

### 3.3. Biofilm Thickness

Due to the variability in biofilm thickness arising from day to day, the data were normalised by biofilm thickness of the control on the given day. All PLS-containing formulations (4, 6, and 8%) then showed significantly reduced normalised biofilm thickness compared to the control (Figure 4). In addition, the formula containing 8% PLS had significantly less biofilm thickness compared to that containing 4% PLS. Additionally, a linear decline in biofilm thickness was observed with increasing PLS concentration (Figure 4).

When biofilm mass and biofilm thickness were both normalised by results for the control, the data within experimental error overlapped. Additionally, both showed a linear decline in normalised mean with increasing PLS concentration (Figure 4). Linear declines of up to 28% and 33% in the biofilm mass and the biofilm thickness, respectively, were observed with increasing PLS concentration to 8% (Figure 4).

### 3.4. Polylysine Release

Percentage PLS release was proportional to the square root of time up to 1, 2, or 3 months with 4%, 6%, and 8% PLS, respectively, but then levelled (Figure 5). The overall general linear model revealed a significant difference in the percentage of PLS released between different composite formulations at every time point. Cumulative release in μg/disc at 24 h and final percentage release both increased with raising the PLS filler content (Figure 6). Within error, final percentage release was 60/350 times the 24 h μg/disc release for all formulations (Figure 6). Final percentage releases were 42%, 28%, and 13% with 8%, 6%, and 4% PLS in the filler phase, respectively. Release at 24 h was 241, 163 and 85 μg/disc, respectively (Figure 6). At 6 h, the release was half these values.

## 4. Discussion

In the current study, the antibacterial effects of composites formulated with simultaneously increasing levels of antibacterial PLS and remineralising MCPM was evaluated. MCPM/PLS weight ratio was fixed at 2. From relative densities and molecular weights, this gave approximately comparable volumes of the two components and the moles of MCPM and the lysine monomer units in all formulations. Previous studies have shown that, in composites, these may work synergistically at promoting surface mineralizing properties [22]. The aim of this study, however, was to assess how formulations with these two components affect *S. mutans* biofilm deposition and survival on the composite surface. Subsequent polylysine release rates were used to explain the large differences observed between materials with different PLS and MCPM levels.

Polylysine is effective against a wide range of bacteria, although concentrations required tend to be much higher than for conventional antibiotics or agents such as chlorhexidine. A recent study showed that increase in PLS concentration above 250 ppm caused reduction in survival rate, enzymatic activity, and adenosine triphosphate (ATP) levels of *Staphylococcus aureus.* It also increased cell collapse and membrane permeability. Additionally, it destroyed the peptidoglycan component of the cell wall and caused changes in the levels of several metabolites [23]. Another study compared the action of PLS of 750 and 74 ppm on Gram-positive and Gram-negative bacteria, *Listeria innocua*, and *E. coli*, respectively. In both cases, PLS exerted its activity by binding to negatively charged phospholipid head groups in the lipid bilayer of bacterial membranes and destabilized them, thus increasing permeability. In the case of *E. coli*, PLS also binds to and destroys the lipopolysaccharide component of the outer membrane [30].

The minimum inhibitory concentration of PLS against *S. mutans* with the standard initial inoculum level of 5.5 × 10^5^ CFU/mL was found to be 20 μg/mL [31], but this increases with the higher bacterial levels seen in the oral cavity. *S. mutans* is a Gram-positive facultative anaerobic bacterium that was first identified as a cariogenic bacterium in 1924 [32]. A previous longitudinal study reported that 96.6% of patients aged between 6–30 have *S. mutans* present in their oral cavity [33]. In a recent study, composite formulations that released 93 ppm PLS upon the first 24 h of water immersion were able to reduce an initial inoculum level of 8 × 10^5^ CFU/mL down to 10^2^ CFU/mL [25]. In this present study, the formulation with 4% PLS with a similar level of PLS release, however, was unable to prevent *S. mutans* biofilm growth. This was likely due to the addition of sucrose enhancing the rate of bacterial growth in addition to enabling production of a protective extracellular matrix (ECM) [34]. The observed linear decline in biofilm mass with increasing PLS suggests it may have been causing a linear decline in the numbers of viable bacteria in the surrounding medium. Alternatively, this effect could have been independent of PLS bactericidal activity and due instead to biofilm matrix disruption caused by PLS [35].

Since CV may stain the ECM in addition to the bacterial cell wall, it provides a measure of the combined live and dead bacteria. In the present study, a live/dead stain was additionally used to determine the fraction of the bacterial population that were live. In regard to the live/dead stain, propidium iodide staining (red) revealed a damaged cell membrane. A statistically significant increase in the percentage of dead *S. mutans* was apparent with the composites containing 6% and 8% PLS compared to the 4% and the control formulations. This study, in conjunction with the previous work [25], suggests that the concentrations of PLS and MCPM in the filler need to be increased significantly to be effective when sucrose is added.

Considerable variation in biofilm thickness detected by confocal microscopy was observed between experiments, although this was not apparent in the biofilm mass measured by CV staining. This may reflect differences in biofilm structural organization between experiments. Nonetheless, when the data were normalised to the control in individual experiments, a consistent decline in both biofilm mass and thickness with increasing PLS concentration was evident.

The most widely used antibacterial agent in dental composites has been chlorhexidine (CHX). Previous studies, however, showed development of antibacterial resistance toward CHX [36] as well as severe [37] and sometimes fatal hypersensitivity reactions [38]. Furthermore, release from hydrophobic composites [39,40] can be restricted and can require addition of hydrophilic components to promote water sorption, which decreases strength. Whilst, in standard tests, minimum inhibitory concentrations of PLS are generally much higher (20 ppm) [31] than those of CHX (<1 ppm), the new studies suggest its very much higher aqueous solubility enables faster release from the composite surface.

In order to quantify the kinetics of PLS release from each formulation, a ninhydrin assay was used. Ninhydrin is used for the detection of ammonia, primary or secondary amines, amino acids, peptides, and proteins [41]. The reaction between NH_2_-group present in protein, peptide, or amino acid with ninhydrin (originally yellow color) produces a coloured ninhydrin chromophore (deep purple colour) (Ruhemann’s purple λ_max_ 570 nm) [41]. PLS is one of the peptides that can be detected by the ninhydrin reaction. Under appropriate conditions, the intensity of the purple colour is proportional to the NH_2_-group (PLS concentration). Ninhydrin reaction is extremely sensitive to low concentrations of protein, peptide, or amino acid [41].

Unlike previous studies [22], there was no initial burst release of PLS in this new study. This may have been a consequence of higher filler content in the earlier work (80 wt%) leading to particles making more direct contact with the surface. In the current study, the percentage of PLS released was instead proportional to the square root of time. This is expected from the Higuchi equation for drug release from thin layers [42].
(2)$$\frac{P_{t}}{P_{\infty }}=\frac{4}{L}\sqrt{\frac{Dt}{\pi }}$$
$P_{t}$ and $P_{\infty }$ are the percentage *PLS release* at time *t* and at infinity, respectively. *D* is a diffusion coefficient and L the thickness of the layer top and bottom of the sample from which the drug is released. If all the PLS is released from this layer but none is released from the remaining material bulk, layer thickness would be expected to be given by:(3)$$L=\frac{P_{\infty }h}{100}$$
where *h* is half the sample thickness and is equal to 0.5 mm. As maximum percentage polylysine releases are 13%, 28%, and 42%, the top and the bottom surface layers depleted of PLS are expected from this expression to be 65, 140 and 210 μm with 4%, 6%, and 8% PLS in the filler, respectively. These are approximately 2, 4, and 6 times the diameters of the PLS and the MCPM particles. The increase in thickness is likely due to higher levels of these hydrophilic components upon their release, generating pores in the surface that enable PLS release from deeper within the specimens. Doubling the concentration of PLS in the filler therefore more than doubles the release from the composite.

Combining Equations (2) and (3) gives:(4)$$P_{t}=\frac{400}{h}\sqrt{\frac{Dt}{\pi }}$$

At 24 h, this becomes:(5)$$P_{24}=\frac{2000}{h}\sqrt{\frac{D}{\pi }}$$

Additionally, from Figure 6:(6)$$P_{24}=100\frac{w_{24}}{w_{c}}=\frac{\left[14+2c\right]}{100mf}$$
*w*_24_ and *w*_c_ are the weight of PLS released at 24 h in micrograms and total PLS in the composite disc. *m*, *c*, and *f* are mass of disc in grams (0.13 g), percentage of PLS in the filler, and filler fraction (0.75) in the composite. This equation shows that, upon doubling the PLS concentration in the filler, the early release in micrograms more than doubles. *h*, *m*, and *f* are all constants. Comparing Equations (5) and (6) therefore suggests that, upon increasing c, the diffusion coefficient for PLS increases, enabling faster release. This may be a consequence of higher c increasing PLS, MCPM, and water sorption. Further studies would be required to separate the effects of PLS and MCPM variables. The results, however, show that doubling both PLS and MCPM filler content more than doubles early PLS release mass.

When the composite is in direct contact with the dentine, release of PLS would be restricted. Release could occur, however, upon composite/dentine interface damage creating a water filled gap. These gaps may occur due poor bonding technique generated through composite shrinkage upon placement or formed at a later time due to cyclic stress on the bond. Rapid release of PLS into this gap could help to prevent biofilm formation initiating recurrent caries. The caries could, at early times after restoration, arise due to residual bacteria, or, if the gap is larger than a few microns, may be initiated by bacteria penetrating from the oral cavity. Additional diffusion of nutrients from the oral cavity would accelerate the bacterial growth within the gap.

If release of PLS is sufficiently fast to enable the minimum inhibitory concentration to be reached in the medium before exponential bacterial growth, then biofilm formation should be prevented. Alternatively, as occurred in this study, if release is not sufficiently fast to prevent biofilm initiation, longer-term release may kill the bacteria that are adherent on the composite surface. This study suggests that higher MCPM and PLS are required to achieve this mechanism of biofilm destruction. This antibacterial action would enable reduced risk with partial caries removal and possibly allow less affected dental hard tissue removal. In addition, it might increase the survival rate of the restoration and decrease the prevalence of recurrent caries. Excessive release of surface PLS, however, would have a negative effect on the top surface roughness and hardness. High levels could additionally reduce mechanical properties. This, however, could be addressed through use of the composite with high levels of MCPM and PLS beneath the control composite. This would likely block release of PLS from the top surface into the oral cavity.

## 5. Conclusions

Composites were prepared with PLS added to the filler phase at levels of 0, 4, 6, and 8 wt% and with monocalcium phosphate added simultaneously at double these levels. Upon incubating set composite discs with *S. mutans* and sucrose, a linear decline in surface biofilm mass and thickness occurred with increasing PLS level. Conversely, an abrupt increase in dead bacteria within the biofilm was observed with PLS levels above 4 wt%. Release studies suggest this was a consequence of 24 h PLS release needing to be above a critical level to kill bacteria within the surface biofilms. Early PLS release from the composite was proportional to the square root of time, as expected for a diffusion controlled process. Release levelled below 100%, suggesting it was from surface layers, which increased in thickness with raising PLS level. Upon damage of the composite restoration/tooth interface, either through polymerization shrinkage during placement or upon cyclic loading, a new composite surface was generated that was in contact with fluid. With rapid and sufficient PLS release from this new surface, risk of recurrent caries within the microgaps formed could potentially be reduced.

## Figures and Tables

**Figure 1 jfb-11-00055-f001:**
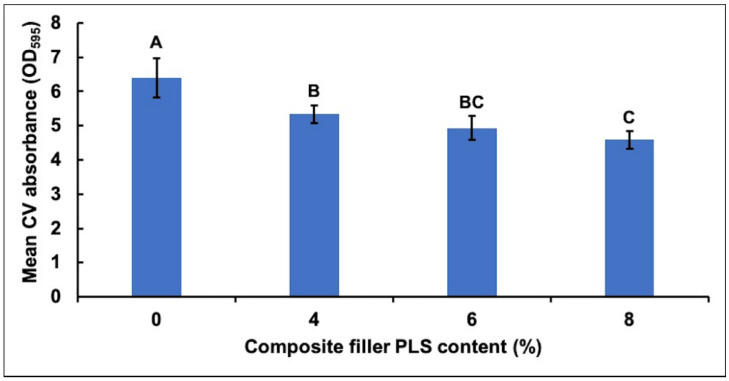
Mean ± (95%CI, *n* = 4) absorbance (OD_595_) due to crystal violet (CV) adsorbed by biofilms on the four composite formulations with 0, 4, 6, or 8 wt% polylysine (PLS) in filler phase (error bars are for 4 repetitions on different days with an average result of 3 specimens per formulation each day). The absorbance values were 10 times those measured in the spectrometer to account for dilution. Composite formulations with the same uppercase letter/s above the bars were not significantly different at *p* < 0.05.

**Figure 2 jfb-11-00055-f002:**
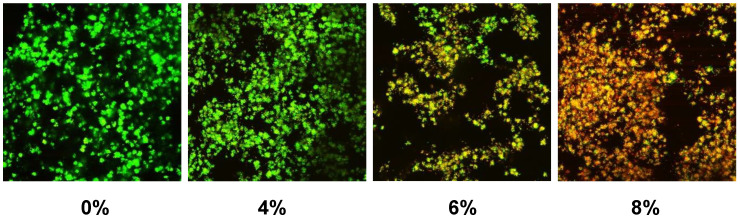
Example confocal microscopy images of live/dead stained biofilms formed on composite formulations containing different percentages of PLS added to the filler phase. Live bacteria stain green whereas dead bacteria are red.

**Figure 3 jfb-11-00055-f003:**
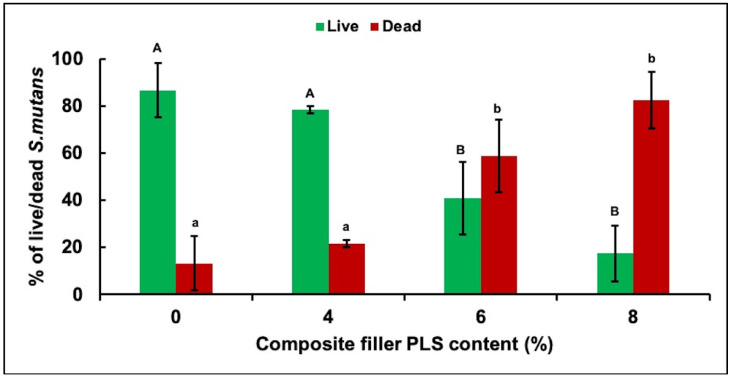
Mean ± (95%CI, *n* = 3) percentage of live and dead *S. mutans* in biofilm formed on discs of the four composite formulations with 0, 4, 6, or 8 wt% PLS in filler phase at 24 h (error bars are for 3 repetitions on different days with 1 specimen per formulation per day). Composite formulations with the same uppercase/lowercase letter/s were not significantly different at *p* < 0.05.

**Figure 4 jfb-11-00055-f004:**
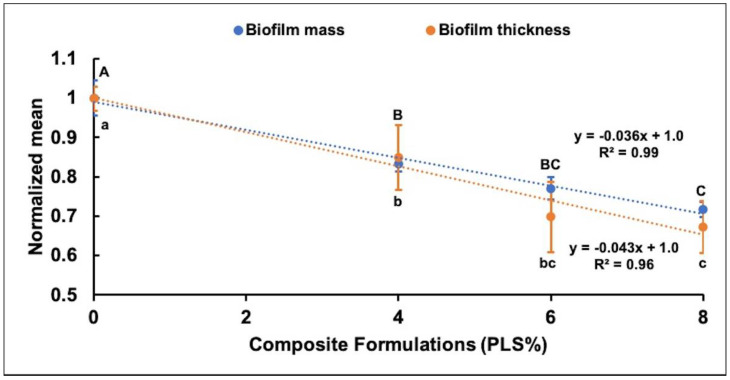
Scatter graph showing the relationship between normalised mean ± (95%CI) biofilm thickness (*n* = 3) and biofilm mass (*n* = 4) (Y-axis) for different composite formulations (X-axis). For biofilm mass and thickness, the formulations with the same uppercase letters above and lowercase letters below the bars, respectively, were not significantly different at *p* < 0.05.

**Figure 5 jfb-11-00055-f005:**
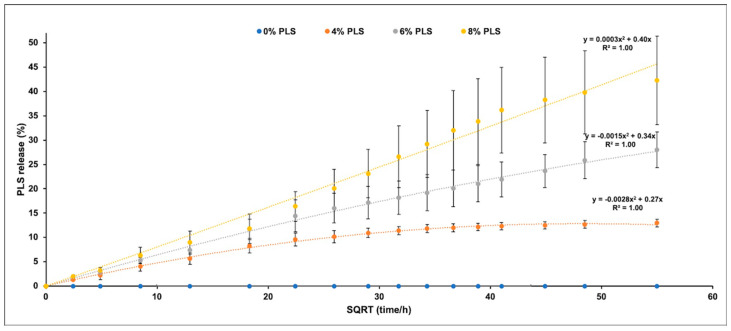
Scatter graph of mean ± (95%CI, *n* = 3) percentage PLS release (Y-axis) versus the square root of time/hours (X-axis) for each composite formula up to 18 weeks (the legend shows the composite filler PLS %). All final release values were significantly different for different PLS levels at *p* < 0.05.

**Figure 6 jfb-11-00055-f006:**
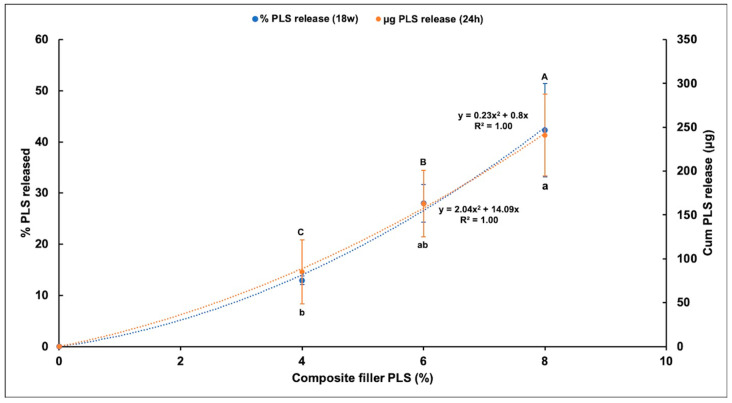
Scatter graph of mean ± (95%CI, *n* = 3) cumulative percentage PLS release at 18 weeks (left *Y*-axis) and mean ± (95%CI) cumulative PLS release in (μg/disc) at 24 h (right *Y*-axis) for different composite formulations (*X*-axis). The formulations with the same uppercase letter/s above or lowercase letter/s below the bars that indicate 18 week and 24 h data, respectively, were not significantly different at *p* < 0.05.

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
