# Peer review of "Effect of Novel Antibacterial Composites on Bacterial Biofilms"

_jfb, 2020, doi:10.3390/jfb11030055_

Round 1
Reviewer 1 Report
This study is well managed, original and very interesting. It show with high performances that dental composites with polylysine (PLS) can decrease S. mutans biofilm.
I suggested that further studies should be performed for these dental composites in vivo validation.
Just one suggestion: The sentence was "The aim of this study" was used two times in the discussion line 252 and 257. This should be changed in my opinion.
Congratulation for your work.
Reviewer 2 Report
The articles describes the preparation of dental composites with PLS and monocalcium phosphate, to adress the problematic of infections in the composite material when used as a restorative after caries. While the aim of the study is accomplished and the results are promising, a careful revision of the writing should be done. Including more linkers will give cohesion to the text and will aid the readers’ understanding of the text.
Some issues need to be adressed before publication:
- In the abstract, I would include an introductory sentence to set the topic of the paper. It seems to be an enumeration of the materials and methods and the results, maybe it could be summarized giving only the essential information.
- Line 66: when do the infections usually appear? Is the release time approppriate in this case?
- Line 131: the fixation is done in methanol? I would write: fixed by exposure to 1mL methanol during 15min.
- Line 133: please provide the information about the crystal violet (supplier, country).
- Line 143: is it 3 time points or 3 independent assays in 3 different days? I can deduce it from the results section, but it could be re-written in order to clarify it.
- Line 168: RSQ is low for a chemical calibration curve, is there any explanation to that?
- Line 192: Do you think that higher concentrations of PLS would provide a better antibacterial effect, or it could have toxic side effects? If so, why not to use a higher concentration?
- Line 199: is the crystal violet absorbance proportional to the number of bacteria? From the confocal images it seems that the number of bacteria is much lower in the concentration of 6%PLS, but this result is not seen in the CV assay. Is it possible that the absorbance values are too high to be in the range of the Lambert-Beer law?
- Line 205: How is the bacteria quantification done in ImageJ? Perhaps a deeper explanation would be helpful in the materials and method section. Moreover, the confocal microscopy image of 0%PLS is too saturated to distiguish between individual bacterium. Is it the original image or it has been modified to enhance it?
- Line 288: the authors state In this study propidium iodide staining reveals a damaged cell membrane. Since this is how the staining works in general, I wouldn’t include in this study at the beginning.
- Line 305: I think it should be its, not it’s
- Line 364: the conclusions are difficult to understand. Consider re-writing to avoid repetition.
- Line 366: I think it’s proportional instead of proportion.
Reviewer 3 Report
critics:
line 25: What is meant by "predominantly dead"? From scientific point of view, no exact information is given. Did you achieve an antimicrobial effect? How would you rank your results? Unclear.
line 173: No information is given in this text section about the number of values and independent experiments. How many values per each condition did you analyse? How many independent experiments were done? Unclear. Section 2.7 must be revised for a better understanding.
Reviewer 4 Report
Authors must be recommanded for their efforts and their intersting research using soundness scientific methods. The referee find scientific merits in the manuscript and in rigorous methodology applied and correctly described. Results are correctly discussed.
Best regards and congratulations.
Round 2
Reviewer 3 Report
The revised version is now improved almost all points and recommendations are fulfilled.
However there is one point which must be revised depending on the editor of chief decision.
From my point of view figures 1 and 3 are not ready for publication, because some values are not really explained within the figure legend. From my point of view, I don't know what is meant by values given in figure 1 on the x-axis.
Furthermore the wording used by the authors within the answer letter to the reviewer is not acceptable: "The antimicrobial effect can be clearly seen in the CLSM images in figure 2". From scientific point of view, such wording isn't scientific due to the point that a validated positive control is missing. Such answer can't be accepted.
Therefore I would recommend to revise the manuscript again.
Author Response
The revised version is now improved almost all points and recommendations are fulfilled.
However there is one point which must be revised depending on the editor of chief decision.
From my point of view figures 1 and 3 are not ready for publication, because some values are not really explained within the figure legend. From my point of view, I don't know what is meant by values given in figure 1 on the x-axis.
Thank you for your comment.
Response: please note that changes were made to Figures 1 and 3 x-axises labels and Figure captions to improve clarity.
Modifications:
In figure 1: the x axis’s title was changed to “Composite filler PLS content (%)”.
In the Caption “ Mean ± (95%CI, n=4) absorbance (OD595) due to CV adsorbed by biofilms on the four composite formulations with 0, 4, 6 or 8 wt% PLS in filler phase (Error bars are for 4 repetitions on different days with an average result of 3 specimens per formulation each day).“
In figure 3: the x axis’s title was changed to “Composite filler PLS content (%)”.
In the Caption “ Mean ± (95%CI, n=3) percentage of live and dead S. mutans in biofilm formed on discs of the four composite formulations with 0, 4, 6 or 8 wt% PLS in filler phase at 24 hours (Error bars are for 3 repetitions on different days with 1 specimen per formulation per day).”
Furthermore the wording used by the authors within the answer letter to the reviewer is not acceptable: "The antimicrobial effect can be clearly seen in the CLSM images in figure 2". From scientific point of view, such wording isn't scientific due to the point that a validated positive control is missing. Such answer can't be accepted.
We apologise for the very poor wording in the original response. This has been modified. We also wish to thank the reviewer for their time and effort in reviewing this paper and comments to improve paper clarity. Please find the new version of the response attached.
